# Enhancer Function in the 3D Genome

**DOI:** 10.3390/genes14061277

**Published:** 2023-06-16

**Authors:** Sergey V. Razin, Sergey V. Ulianov, Olga V. Iarovaia

**Affiliations:** 1Institute of Gene Biology Russian Academy of Sciences, 119334 Moscow, Russia; sergey.v.razin@gmail.com (S.V.R.); sergey.v.ulyanov@gmail.com (S.V.U.); 2Faculty of Biology, M.V. Lomonosov Moscow State University, 119234 Moscow, Russia

**Keywords:** enhancer-promoter communication, 3D genome, chromatin compartment, phase separation

## Abstract

In this review, we consider various aspects of enhancer functioning in the context of the 3D genome. Particular attention is paid to the mechanisms of enhancer-promoter communication and the significance of the spatial juxtaposition of enhancers and promoters in 3D nuclear space. A model of an activator chromatin compartment is substantiated, which provides the possibility of transferring activating factors from an enhancer to a promoter without establishing direct contact between these elements. The mechanisms of selective activation of individual promoters or promoter classes by enhancers are also discussed.

## 1. Introduction

Enhancers were discovered in experiments on the transfection of expression constructs with a reporter gene into eukaryotic cells. It turned out that the inclusion of some fragments of viral genomes in such constructs significantly increases the level of reporter gene expression [1,2,3,4]. DNA elements that increase reporter gene expression have been called enhancers [5]. Subsequent experiments demonstrated that enhancers exhibit their activity regardless of position relative to the promoter (upstream or downstream of the gene) and orientation with respect to the direction of transcription of the reporter gene [6,7,8]. Viral enhancers did not demonstrate pronounced specificity with respect to the promoter. They were able to activate transcription directed by various factors, including many tissue-specific promoters. At the same time, some cellular enhancers significantly better activated the promoters of those genes that were their natural targets [9]. All known enhancers are platforms containing transcription factor binding sites, often clusters of binding sites for several transcription factors [10]. In some cases, transcription factors bound to an enhancer interact with each other, forming a single complex that provides a surface for binding transcription coactivators. Such complexes are called “enhanceosome” [11,12]. However, a more typical situation is when several enhancer-associated transcription factors exhibit additive activity [13]. Removing one of the binding sites or changing their order has little effect on enhancer activity [14]. In some cases, individual enhancers are organized into clusters. A typical example is the Locus Control Region of the vertebrate β-globin gene locus [15,16]. Cluster organization is characteristic of the so-called super-enhancers [17,18]. Tissue specific enhancers contain binding sites for tissue-specific transcription factors [19,20]. 

An analysis of the positions of enhancers in the genome showed that they can be located at a considerable distance (up to 1 Mb or more) from the target promoters. However, most enhancers are located at a relatively short distance (up to 50 kb in mammals and up to 10 kb in Drosophila) from target promoters [21]. In some loci, enhancers and their target genes are interspersed by other genes whose expression is not regulated by these enhancers. This raises the question of mechanisms of enhancer-promoter communication and determinants of enhancer selectivity (see below). According to recent estimates, the number of enhancers in the human genome ranges from 400,000 to 1,000,000 [22,23]. Consequently, each promoter can be regulated by multiple enhancers. It should be noted, however, that the above estimates of the number of enhancers are made by identifying potential enhancers based on the analysis of epigenetic profiles without confirmation of enhancer activity in functional tests. Currently, targeted suppression of enhancer activity in a living cell could be achieved by recruitment of the enzymes that introduce repressive modifications of chromatin [24]. Fusion of the repressive KRAB (Krüppel-associated box) domain with catalytically inactive Cas9 (dCas9) was applied for the identification of 664 functionally significant out of 78,776 potential enhancer-promoter interactions in the human erythroid cell line (K562 cells) [25]. 

Although the number of enhancers in mammalian genomes can be overestimated, activation of a gene or gene locus by multiple enhancers is quite common [26]. In some cases, several enhancers form a functionally interdependent network, so that the exclusion of any single enhancer from this network results in the loss of all enhancer activity [27].

Recent advances in the study of the 3D genome have shown that the spatial organization of the genome plays an important role in controlling promoter activation by enhancers. Key observations were made using experimental procedures based on the proximity ligation protocol [28,29] and known as 3C technologies or C methods [30,31]. All these procedures are based on the cutting of DNA in formaldehyde-fixed cells, followed by ligation of the ends of the resulting DNA fragments. In this case, cross-ligation can occur between DNA fragments that are far from each other on the DNA molecule but reside in physical proximity within the cell nucleus. Analysis of such chimeric DNA fragments makes it possible to judge how often certain parts of the genome are located next to each other. Using these procedures, two important observations were made that are directly related to the regulation of transcription by enhancers. First, it was shown that active enhancers are often in physical proximity to the promoters they activate, regardless of the genomic distances between these enhancers and promoters [32,33,34]. Second, it was demonstrated that eukaryotic chromosomes are partitioned into so-called topologically associated domains (TADs) [35,36,37], which appear to restrict the areas of enhancer action [38,39,40,41]. The distinctive feature of TADs is that the spatial contacts between remote genomic elements occur preferentially within the TAD, whereas inter-TAD contacts are less probable. Therefore, the preferred function of enhancers within TAD may be related to the necessity of establishing spatial contacts with target promoters. In mammals and other vertebrates, TADs are formed by the dynamic extrusion of DNA loops [42,43], mediated by cohesin motors [44,45]. Convergent binding sites for the versatile transcription factor CTCF (CCCTC-binding factor) located at TAD borders [46] block the movement of loop extruders [47,48,49]. 

## 2. The Mechanism of Enhancers Action

Although enhancers were discovered more than 40 years ago and have been actively studied since then, the mechanism of their action remains elusive. In eukaryotic cells, transcription occurs in cycles [50,51,52]. The level of transcription depends on the duration of the cycles and their frequency. Existing experimental data suggest that enhancers increase the frequency of cycles, while their duration is determined by the properties of the promoter [53,54,55,56]. The generally accepted model of enhancer action is that some activator complex is assembled on the enhancer, which is then transferred in one way or another to the target promoter [21]. The activator complex includes transcription factors, chromatin remodeling factors, enzymes that introduce activating chromatin modifications (primarily histone acetylase CBP(CREB-binding protein)/P300 (E1A binding protein p300) as well as a mediator, general transcription factors, and RNA polymerase II. Relatively recently, enhancers have been shown to be sites for the assembly of a complete Pol II preinitiation complex and bidirectional transcription of short-lived enhancer RNA (eRNA) [57,58,59]. Several studies have shown that the level of eRNA transcription correlates with enhancer activity [60,61]. However, the functions of eRNA remain unclear [62,63]. Various observations have led to the proposal of several models that still need further testing [64]. It has been reported that binding of eRNA to CBP stimulates acetyltransferase activity. Possibly, interaction with eRNA can modulate the activity of other epigenetic “writers” or chromatin remodeling complexes. In addition, eRNA can capture various transcription factors [65], thereby increasing their local concentration on the enhancer. This can contribute to the achievement of threshold concentrations of proteins involved in the formation of a phase condensate containing components of the transcription apparatus and various regulatory proteins [66,67]. The results of several studies suggest the involvement of eRNA in the formation of enhancer-promoter loops, although the specific mechanisms remain unclear [68,69].

The ability of enhancers to initiate transcription raises the problem of the functional distinction between enhancers and promoters [70,71]. Initially, enhancers and promoters were considered fundamentally different regulatory elements of the genome. Now, there is convincing evidence that promoters can have enhancer activity [72,73,74,75], while intragenic enhancers can act as alternative promoters [76]. Interestingly, promoters with enhancer activity possess some epigenetic signatures typical for enhancers, such as p300 binding and a high H3K27ac/H3K4me3 ratio [72]. Similar to enhancers, promoters contain nucleosome-free regions (sites of hypersensitivity to DNase I) and transcription factor binding sites. Most of the known transcription factors bind to both enhancers and promoters [70,71]. According to current views, transcription initiation requires the reaching of a certain critical concentration of components of the transcriptional machinery at a promoter. This can be provided by transferring their additional amounts from one or several enhancers. The question is: why cannot the same be achieved by increasing the number of transcription factor binding sites on the promoter? One possible explanation is that the splitting of the regulatory platform into several modules (promoter and remote enhancers) provides more opportunities for regulation. In this regard, enhancer-promoter contact in the nuclear 3D space serves as a mechanism for the assembly of a complete regulatory platform. It should be noted that this regulatory platform can be assembled from several enhancers and can serve to simultaneously activate several promoters. The possibility of constructing different regulatory modules should increase the potential of the regulatory chain, whereas the attraction of several promoters to the same regulatory module can ensure the coordination of the expression of several genes.

## 3. Establishing Communication between Enhancers and Promoters

Regardless of the nature of the factors transferred from the enhancer to the regulated promoter, some transferring channel should exist. The creation of such a channel is usually referred to as “enhancer-promoter communication” (E-P communication). Three main models for E-P communication were proposed: tracking, linking, and looping. The first model assumes that factors initially bound to the enhancer are somehow moved along the chromatin fibril and reach the target promoter. This model is illustrated by the movement of various enzymes that are held in complex with an elongating RNA polymerase that performs intergenic transcription (Figure 1A) [77].

The linking model postulates that certain protein factors bind to the fragment of the chromatin fibril separating the enhancer and promoter and, interacting with each other, provide compression of this fragment. This results in spatial proximity between enhancer and promoter. As an example of this mechanism, oligomerization of the Drosophila architectural protein Chip between the enhancer and the promoter can be mentioned [78,79]. Within the framework of modern ideas about the compartmentalization of the cell nucleus, the linking model can also be represented as the formation of a common phase condensate, including enhancer, promoter, and a separating fragment of a chromatin fibril. Looping of the DNA fragment separating the enhancer and promoter was first proposed to explain the mechanism of action of remote regulatory elements in prokaryotes [80]. As applied to enhancers in vertebrate animals, this model received strong confirmation after the development of methods for studying the spatial organization of the genome using the proximity ligation technique [29]. It has been shown that remote enhancers in the mouse β-globin gene domain form a single complex to which the genes that need to be activated are recruited [32]. Subsequently, similar observations were made on other models [81,82] (for a review, see [83,84,85,86]). The functional significance of the physical approximation of the promoter to the enhancer was demonstrated in experiments on the activation of transcription through the forced formation of enhancer-promoter loops [87,88]. Although the looping model of E-P communication is almost universally accepted, some observations are not consistent with this model. For example, activation of the *Shh* gene during neuronal differentiation of mouse embryonic stem cells is accompanied by a decrease in the spatial proximity between *Shh* and the SBE6 enhancer that controls its work [89]. However, more typical is the juxtaposition of the enhancer and the target promoter during transcriptional activation. A recent analysis of enhancer-promoter loops using the micro-C technique demonstrated that in K562 cells, more than 65% of functionally confirmed enhancer-promoter pairs are characterized by the formation of enhancer-promoter loops [90].

Mechanisms ensuring the retention of enhancers and promoters in spatial proximity remain the subject of discussion. Several models are being considered. The most developed is the model with the participation of architectural elements (insulators) and proteins associated with them. A significant number of architectural proteins are encoded in the Drosophila genome. In model experiments, it was convincingly shown that contacts between these proteins can modulate the spatial organization of the genome, including keeping enhancers close to promoters [84,91,92]. In the mammalian genome, the most important architectural proteins are CTCF and cohesin. Contacts between convergent CTCF binding sites are established by the extrusion of chromatin loops with cohesin [42,46,48,93]. Accordingly, these contacts are dynamic and exist as long as the extrusion process takes place.

CTCF binding sites are often found adjacent to promoters and enhancers. Accordingly, the extrusion of chromatin loops will contribute to the convergence of promoters and enhancers. Binding of CTCF to the recognition motives on DNA can be regulated through DNA methylation and other ways (for example, through competition with other proteins and even with non-coding RNAs such as Jpx (just proximal to XIST) [94]). Changes in CTCF binding profiles seem to play an important role in the reconfiguration of the 3D genome during cell differentiation [95,96]. 

Once spatial contacts between enhancers and promoters are established, they can be maintained by mechanisms that do not rely on DNA loop extrusion. Current results strongly suggest that the demixing of enhancer- and promoter-associated proteins results in the assembly of a phase condensate that keeps the enhancer and promoter in spatial proximity [97,98,99]. Therefore, mechanisms that establish and maintain enhancer-promoter proximity may be fundamentally different (see Section 5 for further discussion). 

## 4. Chromatin Hubs or Chromatin Compartments

The results of the initial studies using C-methods were interpreted in terms of the establishment of direct enhancer-promoter contacts through the formation of protein bridges between enhancers and promoters. It was assumed that the DNA fragments crosslinked by such bridges could be solubilized, after which proximity ligation could be carried out in a dilute solution [29]. However, subsequent studies have shown that spatial proximity between distant genomic elements can only be observed within the folded chromatin fibril [100]. Accordingly, the in situ Hi-C protocol is now commonly used [46]. Although it is generally assumed that the proximity ligation protocol allows for the ligation of only fragments that are in close proximity in the 3D space of the nucleus [29], the mobility of DNA ends within fixed nuclei has not been specifically analyzed. SDS extraction, used in most C-methods, solubilizes a significant portion of histones (Golov et al., unpublished results), which should lead to chromatin decompactization even in the presence of formaldehyde crosslinks. Due to this, DNA ends available for ligation can scan a certain territory, the size of which remains unclear. The results of visualization of enhancers and target promoters in fixed and living cells do not support the assumption that the enhancer is in direct contact with the target promoter. In most of the cases studied, the establishment of communication between the enhancer and the promoter correlates with a reduction in the distance between them. However, this distance still remains significant (100–400 nm [34,101,102,103]. In some cases, the distance between the enhancer and the target promoter does not change upon transcription activation [104] or even increases [89]. Taken together, these observations are most easily interpreted in terms of a model that suggests the assembly of a common activator compartment represented by a small nuclear volume occupied by folded chromatin fiber and associated proteins [105]. The location of enhancer and promoter within this compartment does not necessarily imply direct E-P contact but facilitates the transfer of transcription factors and components of the transcription machinery from the enhancer to the promoter. Several observations suggest that such a compartment may represent a phase condensate [98]. Indeed, some enhancer-promoter loops are destroyed or weakened upon cell treatment with 1,6-hexanediol, an agent that destroys phase condensates [106]. On the other hand, a significant proportion of the EP loops are stable under acute cohesin depletion and therefore are not maintained by extruders. Rather, the loop bases have microcompartment properties [105]. Being located inside the activator compartment, the enhancer can be at different distances from the promoter in individual cells. Only in some cells can the enhancer and promoter be close enough to each other to be captured by proximity ligation. This can explain why even the strongest C-captured contacts are detected in less than 10% of cells [107,108,109].

## 5. Mechanisms of the Target Promoter Search

Despite some uncertainty about how close an enhancer should be to an activated promoter, the loop model for establishing E-P communication is currently the most reasonable. Stimulation of promoter activity by forced looping with an enhancer is a strong argument in favor of this model [87,88,110,111]. In this regard, the question of how the juxtaposition of promoter and enhancer is established seems to be topical. Two models of this process have been proposed: (i) random search followed by fixation of the mutual position of the juxtaposed enhancer and promoter, and (ii) directed movement of the enhancer towards the promoter [112]. The first scenario is supported by the results of model experiments, which demonstrate that the probability of establishing spatial proximity between promoter and enhancer sharply decreases with increasing genomic distance between them [113]. Accordingly, the probability of promoter activation by an enhancer also decreases with increasing genomic distance [113,114], although the relationship is not linear [113]. In the case when the enhancer and the target promoter are located at a considerable distance (along the genome), the establishment of a spatial contact between them because of a random search seems unlikely. In this situation, the DNA looping by the cohesin motor can play an important role in bringing the enhancer closer to the promoter since the search in three dimensions turns into scanning along the chromatin fiber. Loading cohesin onto the enhancer and subsequent unidirectional extrusion of the chromatin loop will move the enhancer along the DNA molecule until it collides with the promoter (Figure 2A). 

Although cohesin extrudes DNA loops symmetrically [115], additional factors (such as the presence of a CTCF binding site near the loading site) can make the extrusion process asymmetric. The presence of so-called stripes or flames on Hi-C maps indicates that unidirectional extrusion from an enhancer is quite common [116,117,118]. If there are other genes between the enhancer and the target promoter, the promoters of these genes must somehow be “skipped” when dragging the enhancer. Another scenario is that the enhancer and promoter have convergently directed CTCF binding sites. In this case, bidirectional extrusion of the loop starting anywhere between the enhancer and the promoter will eventually bring the enhancer closer to the promoter (Figure 2B). If the distance between the enhancer and the promoter is too large, then several extrusion complexes can be loaded, which can organize the segment of the chromatin fibril separating the enhancer and promoter into a rosette of several loops, again bringing the enhancer closer to the promoter (Figure 2C). It is worth saying that the spatial reconfiguration of the genome usually occurs during cell differentiation. After the establishment of spatial communication between the enhancer and the promoter (the formation of a common activator compartment), cohesin is most likely not involved in keeping the enhancer and promoter in spatial proximity. Destruction of cohesin (acute depletion) does not lead to the disappearance of most enhancer-promoter loops and has a minimal effect on the transcription profile [119]. At the same time, cohesin depletion significantly suppresses the activation of inducible genes, especially when the enhancer is located at a significant distance from the promoter [120,121,122]. Consistent with these observations, it was shown in model experiments that cohesin is required to establish/maintain long-range contacts (100 kb) between enhancer and promoter but does not play a role in maintaining contacts at close distances [114]. In Drosophila, the operation of the DNA loop extrusion mechanism has not been demonstrated yet. Classical loop domains (TADs) with a spot at the top of the triangle on Hi-C interaction maps are not observed in Drosophila. It has been proposed that Drosophila TADs are formed by condensation of nucleosomes [123] and thus should be rather called “compartmental domains” [124]. It is possible that the mechanism of chromatin loop extrusion is absent in Drosophila (although acute cohesin degradation has not been performed in Drosophila cells, so we would prefer to refrain from making definitive conclusions). In this case, the establishment of enhancer-promoter contacts occurs by random search. This may be possible due to the smaller genome size and the smaller distance between enhancers and promoters.

It should be noted that the juxtaposition of enhancer and promoter is not always sufficient to activate the promoter. Spatial reconfiguration of the genome usually occurs during cell differentiation and does not always directly lead to the activation of transcription. Enhancers that control the transcription of many inducible genes are often located in close spatial proximity to these genes in the nuclear space, even in the absence of an inducer [81,125]. However, transcription starts only in the presence of an inducer. Consistently, in Drosophila melanogaster embryos, many developmental enhancers establish spatial contacts with their target promoters prior to the beginning of transcription from these promoters [126].

## 6. Selective Activation of Certain Promoters by Remote Enhancers

Enhancers are often located at a considerable distance from target promoters. For example, in the mouse genome, the *Shh* gene and its ZRS (zone of polarizing activity regulatory sequence) enhancer are located about 1 Mb apart [127]. During the “search” of a target promoter (regardless of the “searching mechanism”), enhancers meet other genes that, however, are not activated by this enhancer [21,128,129]. The ability to establish contacts between enhancers and promoters is partially controlled at the level of chromosome organization into topologically associated domains, which in many cases limit the scope of enhancers [38,41,130,131]. However, enhancers possess selectivity towards certain promoters even within a topologically associated domain. This selectivity is provided by a combination of various mechanisms. First, the target promoter must be available. In other words, some transcription factors must already be associated with it, and a preinitiation complex must be assembled on the promoter. Promoters repressed through epigenetic mechanisms cannot communicate with enhancers. Thus, an enhancer that activates the odorant receptor genes establishes communication only with the gene whose promoter is derepressed [132]. Some enhancers show a preference for certain types of promoters. Enhancers were classified according to their ability to preferentially activate promoters containing a DPE (downstream promoter element) or promoters containing a TATA box [133]. There are also enhancers that preferentially activate housekeeping gene promoters. This specificity seems to be determined by the sets of transcription factors that bind to enhancers and different types of promoters [134]. In some cases, the enhancer preferentially activates a particular promoter. A classic example is the promoters of the *dpp*, *Slh*, and *oaf* genes in Drosophila melanogaster. These genes are located within a 70-kilobase segment of the genome. In the same segment of the genome, there are several enhancers that selectively activate only the *dpp* gene promoter [135]. Analysis of a number of enhancer-promoter pairs in the mouse genome has demonstrated that different enhancers show various levels of specificity, ranging from the ability to activate a wide range of promoters to highly specific activation of a particular promoter [136]. The authors suggest that transcription factors associated with enhancers and promoters play a key role in determining the specificity of the enhancer’s action in relation to different promoters. However, this mechanism cannot explain the selective activation of certain promoters in clusters of paralogous genes since the promoters of all genes in a cluster contain binding sites for the same transcription factors. Here, again, various strategies are implemented to ensure selective gene activation [137]. In globin gene clusters, promoters of various genes compete for contact with upstream enhancers (Locus Control Region, LCR); promoters located closer to the LCR along the DNA are preferentially activated [138]. In the natural configuration of the globin gene clusters, these are the promoters of genes encoding embryonic globins. At later developmental stages, these promoters become repressed by epigenetic mechanisms, and promoters of adult globin genes get the opportunity to establish communication with the LCR [139]. In the mammalian protocadherin α (*Pcdhα*) gene cluster, each promoter contains a CTCF binding site capable of looping with a convergent CTCF binding site on the enhancer. However, the CTCF sites on the promoters are methylated, preventing CTCF binding and looping. Selective demethylation of the CTCF binding site on a particular promoter allows that promoter to establish contact with the enhancer, resulting in activation of the expression of that particular *Pcdhα* gene [140,141].

## 7. Perspectives

Cell identity in multicellular organisms is determined by transcription profiles, which are largely controlled by enhancers. Although enhancers were discovered about 40 years ago and have been actively studied since then, many questions regarding both the mechanism of enhancer action and the mechanisms that ensure the specificity of activation of certain promoters remain open. One of the common problems in studying the regulatory systems of the eukaryotic genome is that most of the observations are made in experiments with cell populations. At present, a whole range of experimental approaches have emerged that allow studying individual cells, which should significantly expand our knowledge of the mechanisms of transcription regulation. Currently, single-cell Hi-C studies confirm the presence of TADs in individual cells but demonstrate significant variability in their shape (i.e., in the folding of DNA within TADs) [142,143,144,145]. It was also found that only a portion of TAD borders are shared between most individual cells, whereas other borders appear to be variable [142,143]. The main problem with single-cell Hi-C experiments is that currently generated single-cell Hi-C matrices are sparse. The best single-cell Hi-C map resolution reported so far is 10 Kb [143]. Sparse Hi-C matrices are not suitable for loop calling by commonly used algorithms. However, more sophisticated bioinformatic tools along with modeling approaches [146,147,148] may solve this problem. Microscopic approaches always provide information about individual cells. A recently developed oligopainting strategy [149] combined with super-resolution microscopy allows the path of the chromatin fibril to be traced at relatively long genomic loci [150,151]. The results obtained using this technology confirmed the presence of TADs and demonstrated the variability of their shape [150], which is consistent with the single-cell Hi-C data. In the future, all these approaches can be used to study the functional impact of the variability in the spatial organization of the genome. The outstanding questions to be addressed include, but are not limited to, the following: Does the cell simply tolerate the variability of the three-dimensional genome, or is this variability used to adapt to changing environments by sorting out possibilities and fixing the most appropriate in the current situation? How often can occasional changes in the 3D genome lead to the activation of harmful genes, such as oncogenes? Is it possible to change the transcription profile through mechanical influences on the cell nucleus? Can the presence of an extra chromosome in aneuploid cells affect the transcription profile of genes on other chromosomes through 3D genome reconfiguration? To answer all these questions, strategies for modifying the 3D genome may also be useful. We have already discussed in previous sections the possibility of gene activation by forced formation of enhancer-promoter loops [87,88,110,111]. Manipulation with CTCF binding sites (deletion, inversion, and insertion) is also an obvious direction for 3D genome editing [152,153,154]. Another approach is tethering to selected genomic positions of CTCF fused with dCas9 [155,156]. It can be hoped that, in addition to the questions listed above, the studies using these approaches will finally resolve the issue of the importance of spatial juxtaposition of enhancers and promoters for transcription activation.

## Figures and Tables

**Figure 1 genes-14-01277-f001:**
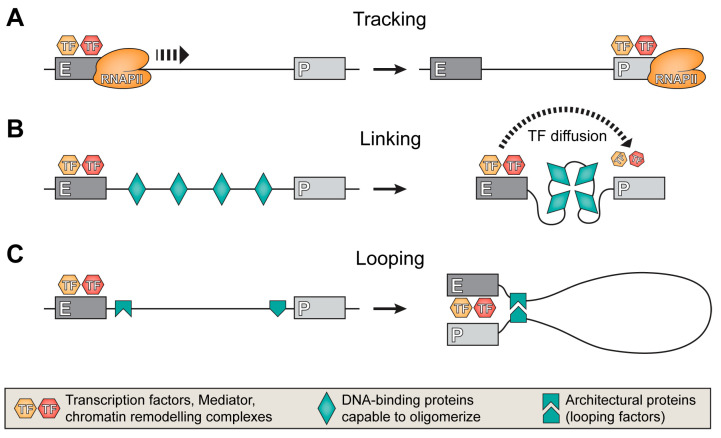
Different mechanisms for establishing enhancer-promoter communication. (**A**) Transcription factors and components of transcriptional machinery gathered on the enhancer are transferred to the nearby promoter by RNA polymerase II, which performs low-level intergenic transcription. (**B**) The condensation of proteins bound to DNA between the enhancer and the promoter brings the enhancer closer to the promoter in physical space. (**C**) The interaction of architectural proteins bound to the enhancer and the promoter leads to the formation of a DNA loop and the juxtaposition of the enhancer and the promoter.

**Figure 2 genes-14-01277-f002:**
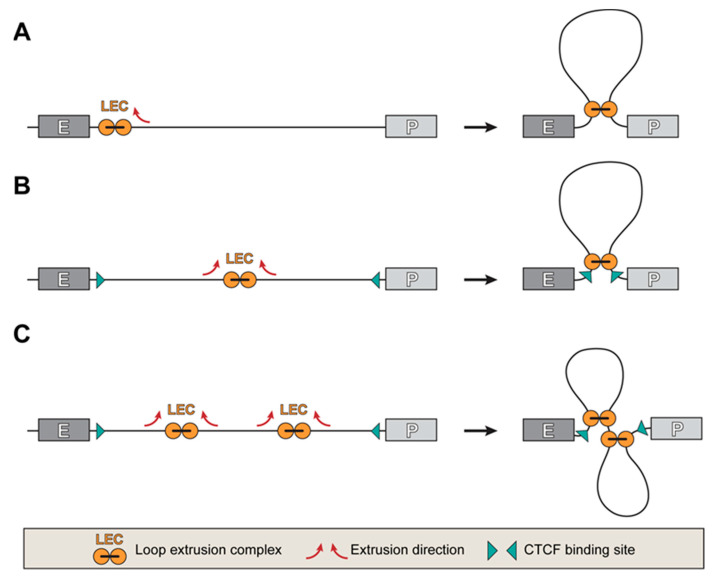
Models illustrating the possibility of approaching the enhancer and promoter through the operation of extrusion complexes. (**A**) Unidirectional extrusion of the DNA loop by the cohesin complex fixed on the enhancer leads to the movement of the enhancer along the DNA molecule up to the meeting with the target promoter. (**B**) Bidirectional extrusion of the DNA loop by a cohesin complex loaded between the enhancer and the promoter results in the juxtaposition of the enhancer and the promoter carrying convergent CTCF binding sites. (**C**) Loading of multiple extrusion complexes between the enhancer and the promoter results in the formation of a rosette of DNA loops and the juxtaposition of the enhancer and the promoter carrying convergent CTCF binding sites.

## Data Availability

Not applicable.

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
