# Peer review of "Enhancer Function in the 3D Genome"

_genes, 2023, doi:10.3390/genes14061277_

Round 1

Reviewer 1 Report

The authors have reviewed the role of enhancers in transcription and outline the various ways in which they are thought to interact with promoters, especially in the context of the 3D genome organization.

While this review is comprehensively written, a lot of the information covered here was also covered in the 2019 Nature collections on the 3D genome, particularly an article by Shoenfelder and Fraser and therefore lacks some novelty. In the concludng remarks the authors comment on newer developments in the field such as studying the 3D genome in single cells and various genome engineering techniques which show promise in understanding E-P communication or enhancer mechanisms. Rather than just alluding to these new methods, I recommend that the authors devote a full section to the findings, advantages, and shortcomings of these methods. If length is a concern then I would recommend that some of the earlier sections be shortened.

I do like the way the article is structured, however, there are key aspects of genome organization such as methods for measuring contacts such as Hi-C and the existence of topologically associated domains (TADs) which are fundamental and therefore in my opinion should be present in the introduction rather than introducing them in various sections halfway through the paper. 

Overall, I think this is an important topic, I just recommend that the authors rework some of the text to improve the narrative and introduce new developments in the field from the last 3-4 years.

There were several instances in the paper where grammar could be improved and typos could be corrected e.g. "this rises the question" on line 46, "their mechanism of their action" on line 64, and "lo-lewel" on line 158. There are several such instances and some other awkward phrases that the authors should pay close attention to. 

More egregiously, there is an entire paragraph which has been repeated, lines 144-154 and 164-174. 

Author Response

Comments and Suggestions for Authors

The authors have reviewed the role of enhancers in transcription and outline the various ways in which they are thought to interact with promoters, especially in the context of the 3D genome organization.

While this review is comprehensively written, a lot of the information covered here was also covered in the 2019 Nature collections on the 3D genome, particularly an article by Shoenfelder and Fraser and therefore lacks some novelty. In the concludng remarks the authors comment on newer developments in the field such as studying the 3D genome in single cells and various genome engineering techniques which show promise in understanding E-P communication or enhancer mechanisms. Rather than just alluding to these new methods, I recommend that the authors devote a full section to the findings, advantages, and shortcomings of these methods. If length is a concern then I would recommend that some of the earlier sections be shortened.

I do like the way the article is structured, however, there are key aspects of genome organization such as methods for measuring contacts such as Hi-C and the existence of topologically associated domains (TADs) which are fundamental and therefore in my opinion should be present in the introduction rather than introducing them in various sections halfway through the paper. 

Overall, I think this is an important topic, I just recommend that the authors rework some of the text to improve the narrative and introduce new developments in the field from the last 3-4 years.

Comments on the Quality of English Language

There were several instances in the paper where grammar could be improved and typos could be corrected e.g. "this rises the question" on line 46, "their mechanism of their action" on line 64, and "lo-lewel" on line 158. There are several such instances and some other awkward phrases that the authors should pay close attention to. 

More egregiously, there is an entire paragraph which has been repeated, lines 144-154 and 164-174. 

REPLY:

At the suggestion of the reviewer, we expanded the introduction to include a brief description of the proximity ligation procedure and a description of the main features of the organization of the 3D genome, which, apparently, are directly related to the regulation of enhancer activity. We have also expanded the final section (currently titled "Perspectives"). We briefly describe the results of 3D genome studies in single cells (both using Hi-C and FISH based approaches), formulate outstanding questions for future research, and outline strategies for 3D genome editing.

Also we have corrected all mistakes in the text mentioned by the reviewer.

Reviewer 2 Report

the review entitled "Enhancer function in the 3D genome" by Razin et al., is an interesting read, though the manuscript has a promising title and a promising start it fizzles out as the paper progresses. The authors have compiled the write-up well however, there are many missed opportunities to analyze and describe the features of the enhancer region which would be conserved or be the guiding force for the entire process. The overall writing style of the paper is detailed but there is little to no inference, the authors have attempted to talk about the various models and possible mechanism of interaction but they just seem to be mentioned and do not have any structure or story to the write up. topics keep coming one after the other. the authors assume that the reader would be expert at these topics and do not require any introduction to the topic they are beginning, similarly the topics do not have conclusions either. The paragraph begins and there is information, followed by information but no inference or commentary whatsoever. I think this is a good review but should be rewritten with a purpose or a storyline and not a giant disjointed essay on the spatial orientation of enhancers. 

major comments

line 52 - 54 references missing

line 86 - 88 the author mentions this fact, it is missed opportunity to show what are the elements of promoter show enhancer activity

line 128 -130, 133 what eukaryotic model are the authors referring to? unicellular/multicellular, vertebrate/invertebrate, please explain.

line 136 - 140, statement appears to be placed randomly, it does not seem to fit the context. 

line 183, what are "C-methods" some preface will be useful for the genearl readers

line 189 - 199, references should be added appropriately

In some places, the syntax does need changes, and the sentences are bulky or do not convey the intended meaning. e.g.

line 200 - 201 rewrite statement

line 34 has unclear syntax

line 46 grammar check required

the paper has some punctuation errors which need attention

Author Response

Comments and Suggestions for Authors

the review entitled "Enhancer function in the 3D genome" by Razin et al., is an interesting read, though the manuscript has a promising title and a promising start it fizzles out as the paper progresses. The authors have compiled the write-up well however, there are many missed opportunities to analyze and describe the features of the enhancer region which would be conserved or be the guiding force for the entire process. The overall writing style of the paper is detailed but there is little to no inference, the authors have attempted to talk about the various models and possible mechanism of interaction but they just seem to be mentioned and do not have any structure or story to the write up. topics keep coming one after the other. the authors assume that the reader would be expert at these topics and do not require any introduction to the topic they are beginning, similarly the topics do not have conclusions either. The paragraph begins and there is information, followed by information but no inference or commentary whatsoever. I think this is a good review but should be rewritten with a purpose or a storyline and not a giant disjointed essay on the spatial orientation of enhancers. 

REPLY

We thank the reviewer for constructive comments. We have expanded Introduction and concluding section (now entitled "Perspectives"). We have also added summarizing sentences to sections 2 and 3 and added several sentences at the end of section 4. We hope that the logic of our discussion becomes clearer in the revised version of the MS.

COMMENTS:

major comments

line 52 - 54 references missing

REPLY:

The necessary references were included (Refs....)

COMMENTS:

line 86 - 88 the author mentions this fact, it is missed opportunity to show what are the elements of promoter show enhancer activity

REPLY:

The authors of the cited articles identified a number of promoters which possesses enhancer activity. It is hardly useful to discuss here all these promoters. However, we have expanded the text by mentioning that promoters possessing enhancer activity are characterized by specific epigenetic signatures: "Interestingly, promoters with enhancer activity possess some epigenetic signatures typical for enhancers, such as p300 binding and high H3K27ac/H3K4me3 ratio {Dao, 2017 #10690}".

COMMENTS:

line 128 -130, 133 what eukaryotic model are the authors referring to? unicellular/multicellular, vertebrate/invertebrate, please explain.

REPLY:

In the revised MS we modified the text in the following way: " As applied to enhancers of vertebrate animals, this model received strong confirmation after the development of methods for studying the spatial organization of the genome using the proximity ligation technique"

COMMENTS:

line 136 - 140, statement appears to be placed randomly, it does not seem to fit the context. 

REPLY:

The text was modified: "Although the looping model of the E-P communication is almost commonly accepted, some observations are not consistent with this model. For example, activation of the Shh gene during neuronal differentiation of mouse embryonic stem cells is accompanied by a decrease in the spatial proximity between Shh and the SBE6 enhancer that controls its work"

COMMENTS:

line 183, what are "C-methods" some preface will be useful for the genearl readers

REPLY:

In the revised version of the MS the description of C-methods is presented in Introduction

COMMENTS:

line 189 - 199, references should be added appropriately

REPLY:

This section is mainly based on our own observations and is aimed to raise questions rather than to give answers. That is why we can only mention our unpublished data.

Comments on the Quality of English Language

In some places, the syntax does need changes, and the sentences are bulky or do not convey the intended meaning. e.g.

COMMENTS:

line 200 - 201 rewrite statement

REPLY:

The statement has been rewritten: In some cases, the distance between the enhancer and the target promoter does not change upon transcription activation {Alexander, 2019 #10686}, or even increases {Benabdallah, 2019 #10677}.

COMMENTS:

line 34 has unclear syntax

REPLY:

The text has been modified: However, a more typical situation is when several enhancer-associated transcription factors exhibit additive activity {Arnosti, 2005 #10667}. Removing one of the binding sites or changing their order has little effect on enhancer activity {Grosveld, 2021 #10448}

COMMENTS:

line 46 grammar check required:

REPLY:

The tipo mentioned by the reviewer was corrected.

COMMENTS:

the paper has some punctuation errors which need attention

REPLY:

The text was edited

Reviewer 3 Report

In my opinion, the review is well-written and covers most of the topics regarding enhancers and promoters. I would like to raise a few questions and comments.

  1. The authors briefly touch on the dynamical component in enhancer-promoter interaction, stating that the loop model is favorable and E-P contacts are observed in less than 10%. However, the authors skip the discussion of 2-3-4 state models. It might be another section, though.

  2. It appears that paragraph (lines 144-154) is duplicated; please refer to lines 164-174.

  3. The sentence "Indeed, the enhancer-promoter loops are destroyed or weakened upon cells treatment with 1,6-hexanediol, an agent that destroys phase condensates [86]." should be softened.

  4. In section 5, the authors speculate on the possible interplay between E-P communication and loop extrusion. It has been shown that the depletion of RAD21 (a subunit of cohesin) has a mild impact on a very small fraction of genes. Also, in this section, the authors didn't comment on species where cohesin is not responsible for loops and TADs, such as Drosophila.

I couldn't find some papers that authors might want to include:

[1] Goel, Viraat Y., Miles K. Huseyin, and Anders S. Hansen. "Region Capture Micro-C reveals coalescence of enhancers and promoters into nested microcompartments." Nature Genetics (2023): 1-9.

[2] Brueckner, David B., et al. "Stochastic motion and transcriptional dynamics of distal enhancer-promoter pairs on a compacted chromosome." bioRxiv (2023): 2023-01.

I found that the quality is fine but I encourage the authors to reread and fix minor inconsistencies.

Author Response

Comments and Suggestions for Authors

In my opinion, the review is well-written and covers most of the topics regarding enhancers and promoters. I would like to raise a few questions and comments.

COMMENT

The authors briefly touch on the dynamical component in enhancer-promoter interaction, stating that the loop model is favorable and E-P contacts are observed in less than 10%. However, the authors skip the discussion of 2-3-4 state models. It might be another section, though.

REPLY

The discussiuon of these models is out of scope of our review due to obvious lack of place in our review.

COMMENT

It appears that paragraph (lines 144-154) is duplicated; please refer to lines 164-174.      

REPLY

This duplication is corrected

COMMENT

The sentence "Indeed, the enhancer-promoter loops are destroyed or weakened upon cells treatment with 1,6-hexanediol, an agent that destroys phase condensates [86]." should be softened.                                                                                                                                 

REPLY

The statement is softened as follows: "Indeed, some enhancer-promoter loops are destroyed or weakened upon cells treatment with 1,6-hexanediol..."

COMMENT

In section 5, the authors speculate on the possible interplay between E-P communication and loop extrusion. It has been shown that the depletion of RAD21 (a subunit of cohesin) has a mild impact on a very small fraction of genes. Also, in this section, the authors didn't comment on species where cohesin is not responsible for loops and TADs, such as Drosophila.                                      

REPLY

The experiments on acute depletion of cohesin are discussed in section  5: "Destruction of cohesin (acute depletion) does not lead to the disappearance of most enhancer-promoter loops and has a minimal effect on the transcription profile {Hsieh, 2022 #10692}".In the revised version of the MS we have added several comments on 3D genome organization in Drosophila: " In Drosophila, operation of DNA loop extrusion mechanism has not been demonstrated yet. Classical loop domains (TADs) with a spot at the top of triangle at Hi-C interaction maps are not observed in Drosophila. It has been proposed that Drosophila TADs are formed by condensation of nucleosomes {Ulianov, 2016 #7313} and thus should be rather called "compartmental domains" {Rowley, 2018 #8070}. It is possible that the mechanism of chromatin loops extrusion is absent in Drosophila (although acute cohesin degradation has not been performed in Drosophila cells, so we would prefer to refrain from making definitive conclusions). In this case, the establishment of enhancer-promoter contacts occurs by random search. This may be possible due to the smaller genome size and the smaller distance between enhancers and promoters."

COMMENT

I couldn't find some papers that authors might want to include:

[1] Goel, Viraat Y., Miles K. Huseyin, and Anders S. Hansen. "Region Capture Micro-C reveals coalescence of enhancers and promoters into nested microcompartments." Nature Genetics (2023): 1-9.

[2] Brueckner, David B., et al. "Stochastic motion and transcriptional dynamics of distal enhancer-promoter pairs on a compacted chromosome." bioRxiv (2023): 2023-01.

REPLY

We have cited the article by Goel at al mentioned by the reviewer.

COMMENT

Comments on the Quality of English Language

I found that the quality is fine but I encourage the authors to reread and fix minor inconsistencies.

REPLY

Several mistakes in the text have been corrected

Round 2

Reviewer 1 Report

The authors have edited the manuscript according to my comments and I recommend it for publication

There may be a glitch on my end but the entire first section (lines 20-71) seem to be duplicated (to lines 72 - 120)

The word sparse is misspelled twice on line 360